# Time-Series of Vegetation Indices (VNIR/SWIR) Derived from Sentinel-2 (A/B) to Assess Turgor Pressure in Kiwifruit

**Alberto Jopia** [1,†] **, Francisco Zambrano** [1,*,†] **, Waldo Pérez-Martínez** [2] **, Paulina Vidal-Páez** [2] **, Julio Molina** [3] **and Felipe de la Hoz Mardones** [4]

1    Hémera Centro de Observación de la Tierra, Escuela de Agronomía, Facultad de Ciencias, Universidad Mayor, Santiago 8580745, Chile; alberto.jopia@ug.uchile.cl
2    Hémera Centro de Observación de la Tierra, Escuela de Ingeniería Forestal, Facultad de Ciencias, Universidad Mayor, Santiago 8580745, Chile; waldo.perez@umayor.cl (W.P.-M.); paulina.vidal@umayor.cl (P.V.-P.)
3    Escuela de Agronomía, Facultad de Ciencias, Universidad Mayor, Santiago 8580745, Chile; julio.molina@umayor.cl
4    Centro Especializado de Riego, Liceo Agrícola El Carmen–SNA Educa, San Fernando 3070000, Chile; felipedelahoz@udec.cl
*    Correspondence: francisco.zambrano@umayor.cl
†    These authors contributed equally to this work.

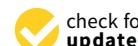

**Simple Summary:** Measures of plant water status on kiwifruit were evaluated using Sentinel-2 (A/B) for two seasons in Central Chile.

**Abstract:** For more than ten years, Central Chile has faced drought conditions, which impact crop production and quality, increasing food security risk. Under this scenario, implementing management practices that allow increasing water use efficiency is urgent. The study was carried out on kiwifruit trees, located in the O'Higgins region, Chile for season 2018–2019 and 2019–2020. We evaluate the time-series of nine vegetation indices in the VNIR and SWIR regions derived from Sentinel-2 (A/B) satellites to establish how much variability in the canopy water status there was. Over the study's site, eleven sensors were installed in five trees, which continuously measured the leaf's turgor pressure (Yara Water-Sensor). A strong Spearman's ($\rho$) correlation between turgor pressure and vegetation indices was obtained, having $-0.88$ with EVI and $-0.81$ with GVMI for season 2018–2019, and lower correlation for season 2019–2020, reaching $-0.65$ with Rededge1 and $-0.66$ with EVI. However, the NIR range's indices were influenced by the vegetative development of the crop rather than its water status. The red-edge showed better performance as the vegetative growth did not affect it. It is necessary to expand the study to consider higher variability in kiwifruit's water conditions and incorporate the sensitivity of different wavelengths.

**Keywords:** turgor; Sentinel-2; vegetation spectral indices; kiwi; SWIR/NIR; time-series

---

## 1. Introduction

In the current climate change scenario, food security and water availability are two of the main challenges facing agriculture [1]. This makes it necessary to adopt strategies that allow for climate-smart agriculture and thus efficient use of water [2]. The change in precipitation patterns due to climate change in Chile has caused a prolonged drought called mega-drought [3]. Its beginning dates back to 2007 when much of the Chilean territory was declared in agricultural emergency due to

drought [4–6]. The prolonged decrease in precipitation has affected the entire hydrological system in central Chile, depleting water reserves in mountains, aquifers, reservoirs, lakes and lagoons [7]. The analysis of the vegetative state in central Chile shows that during the 2019–2020 season it is facing an extreme anomaly with respect to its normal condition (2000–2019), particularly critical for the cropland and forest land cover types [8], which is affecting the production and quality of agricultural crops. Today it is urgent to implement agricultural management practices that allow for the efficient use of water to face the new water scenario.

As defined by Kirkham [9], water use efficiency (WUE) corresponds to the amount of dry matter produced for each millimeter of water that the crop uses through evapotranspiration (ET). The yield of a crop is regulated by its ET rate [10], which in terms of efficiency must be kept at a minimum optimal value that does not affect the biomass production of the crop, as below this threshold, the crop will be under stress. This threshold can be identified by measuring different components within the soil-plant-atmosphere continuum (SPAC) [11]. In the countryside, the most common practice is to measure the water potential of the leaf using the Scholander camera [12]. However, it is a destructive method that requires time and field work, which limits synoptic measurement in space. Different authors have monitored the water statuses of the plant, focusing on the turgor pressure of the leaf [13–17], which corresponds to the pressure generated by water within the cell that compresses the plasma membrane against the cell wall [18,19]. This force decreases during the day due to the loss of water through evapotranspiration and is recovered at night. Zimmermann et al. [20] developed a patch clamp sensor that continuously transmits turgor variation. Specifically, this technique measures the attenuation of the clamping pressure applied externally on the sheet by the patch clamp sensor, called patch pressure (Pp) [21]. The limitation of this method is the need to have enough sensors to be able to make a spatial characterization of the turgor variation in the field.

The water status of crops can be evaluated using optical, thermal or microwave reflectance. Optical reflectance has been widely used due to water absorption characteristics in the spectral range 400–2500 nm [22,23]. However, canopy reflectance is also influenced by factors such as the optical properties of the leaf, the geometries of observation of the sun, the reflectance of the soil, the absorption of atmospheric water and, most importantly, the way in which the leaves are organized on the tree [24–26]. The inversion of a radiative transfer model (RTM) provides one approach [22,27]; however, it requires knowledge of the parameters of the biochemical content of the leaf, which is generally time consuming. A spectral index, through the combination of reflectance at different wavelengths, has the potential to be used in regional scale applications using an empirical approach. The first applications on vegetation have been based on recovering reflectance in the visible and near infrared range (VNIR). Indices based on these regions have been used to assess the health and photosynthetic vigor of vegetation [28–31]. Examples of these are the Normalized Difference Vegetation Index (NDVI) [32] and the Enhanced Vegetation Index (EVI) [33], which incorporate the blue band into the NDVI to reduce the saturation that occurs on surfaces with high albedo values and by the dispersion of the atmosphere [33]. On the other hand, short-wave infrared reflectance (SWIR) is the most sensitive to the variation in vegetation water content due to the higher absorption coefficient [34–37]. It is typically combined with near infrared (NIR) reflectance to mitigate the impact of other factors, such as canopy structure. Some representative indices of the water status of the vegetation have been proposed. Hardisky et al. [38] first developed the Normalized Difference Infrared Index (NDII) to evaluate the spetral respect of *Spartina alterniflora* to different physiological changes, determining that the wavelength between 1.50–1.75 µm presented the best response to the differences in the water content of the leaf. Gao [39] proposed the Normalized Difference Water Index (NDWI) using two channels centered near 0.86 µm and 1.24 µm and determined that NDWI increases as the layers of sheets increase, indicating that NDWI is sensitive to the total amount of water in stacked sheets. Fensholt and Sandholt [40] developed the Shortwave Infrared Water Stress Index (SIWSI) using MODIS band 2 (841–876 nm) and band 6 (1630–1650 nm). Xiao et al. [41] referred to the NDII as the SPOT-4 satellite-based land surface water index. The new multispectral sensor (MSI) attached to the

Sentinel-2 (A/B) satellites provides opportunities for vegetation monitoring due to its higher spatial resolution (10/20/60 m), a spectral range of 13 bands from 443 to 2190 nm, and a revisit capable of covering the Earth every five days [42], allowing the use of high frequent time-series of satellite data for environmental research. Some examples are the use of Google Earth Engine (GEE) and Sentinel-2 to mapping forestry vegetation [43], the use of Landsat data to monitor ecosystem change [44], and for the mapping of bamboo throughout GEE [45]. Although simulation studies [46] have been carried out to evaluate Sentinel-2 data's usefulness to estimate the leaf area index (LAI) and chlorophyll, its ability to assess the water content of the crop it has been less investigated. The objective of this study is to evaluate the use of time-series of spectral indices derived from Sentinel-2 images, obtained during the seasons 2018–2019 and 2019–2020 to detect turgor pressure in kiwifruit trees. Different vegetative indicators obtained from wavelengths in the VNIR and SWIR and their relationship with measurements of foliar turgor will be analyzed.

## 2. Materials and Methods

### 2.1. Study Area

El Carmen agricultural high school is located in the commune of San Fernando, O'Higgins región, Chile (Figure 1). The species studied was kiwi (*Actinidia deliciosa*) Hayward variety, cultivated in a loamy textured soil. The orchard under study was planted in 2009 and has an area of 6.5 ha, divided into two barracks of 3 ha each with 3 irrigation subunits, with a 4.5 m × 3 m planting frame, in a conduction system of New Zealand vineyard and a localized irrigation system with 36 lt/hr micro-sprinkler. This area has a Mediterranean climate (Csb) [47] with moderate rainfall, and an annual average of 383 mm/year in the last 10 years, concentrated in winter, with a prolonged dry season of 7 to 8 months [48]. In addition, in this fruit tree, agricultural management was carried out to enhance the yield and quality of the fruit, from the four stages of the phenological development of the tree (Table 1 and Figure 2). However, irrigation was applied, supplying enough water to satisfy the water demand of the crops, which was monitored by crop evapotranspiration [49], recovered from a meteorological station located 700 m from the orchard. The irrigation supply had variations mainly focused on enhancing the development of the fruit. During E1 and E2 a greater and more frequent amount was applied than in E3 and E4. In these last stages, irrigation events were applied less frequently and with a longer duration to avoid the development of new foliage that would compete for nutrients with the development and maturation of the fruit [50]. The application of the water supply in each season can be observed by recording the soil moisture at different depths using a Sentek probe installed in the orchard (Figure 2).

**Table 1.** Description and approximate start dates of the main stages of agricultural management of Kiwi var. Hayward. Source: Modified from Sabaini and Goecke [50], Sabaini [51].

| Management Stage | Start | End | Description |
|---|---|---|---|
| E1: Load regulation | 15 September | 15 November | It begins about 10 days before budding with the start of floral differentiation. Flower buds develop until flowering. Pruning and thinning work is carried out |
| E2: Pollination | 15 November | 7 January | It begins with the pollination of the flower. The fruit develops and grows at high rates reaching 50% of its weight and 70% of its final volume. In this stage the vegetation reaches its highest water demand and the temperature reaches its maximum |

**Table 1.** *Cont.*

| Management Stage | Start | End | Description |
|---|---|---|---|
| E3: Vegetation management | 7 January | 15 March | Green pruning is carried out, corresponding to the removal of shoots and foliage in order to improve the distribution and penetration of light in the crown |
| E4: Fruit harvest | 15 March | 15 April | The maturity and harvest of fruit is ensured, it is the shortest stage and extends from the third week of March to mid-April. |

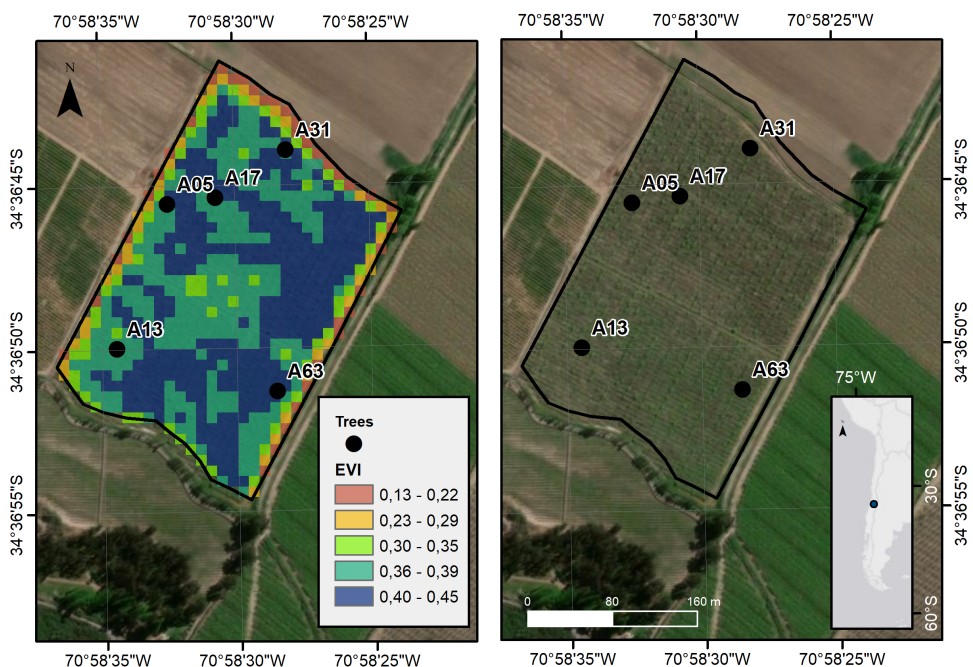

**Figure 1.** Study area of Kiwifruit orchard on El Carmen agricultural high school, San Fernando, Chile.

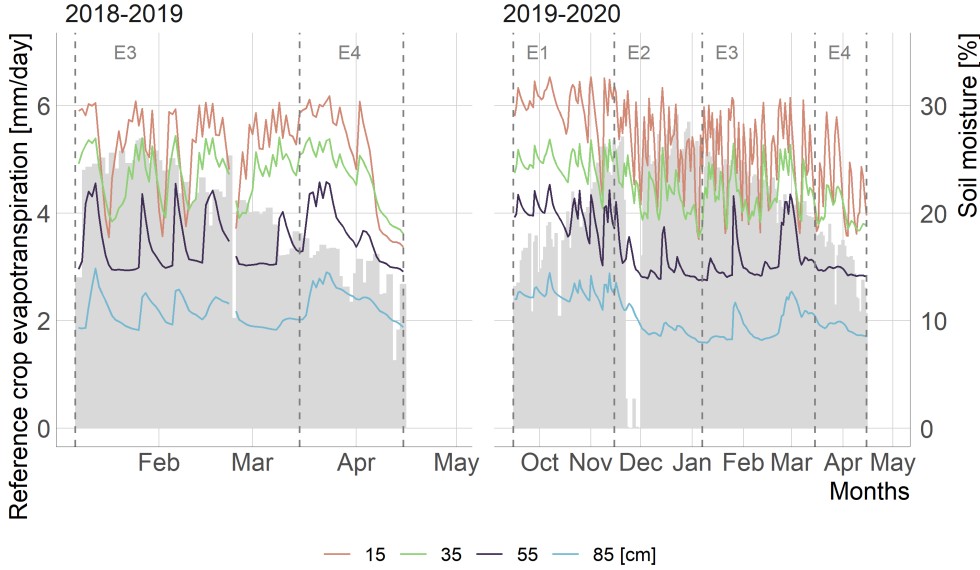

**Figure 2.** Variation of crop evapotranspiration and soil moisture content at different depths (15, 35, 55, and 85 cm) in the kiwi orchard for the 2018–2019 and 2019–2020 season. Horizontal lines correspond to the depths of soil moisture of the Sentek probe. Blank space between late April and early October corresponds to the period without irrigation.

*2.2. Data*

Sentinel-2

The Sentinel-2 mission of the European Space Agency (ESA) is composed of two twin satellites, A and B, which have a multispectral sensor with 13 spectral bands at a spatial resolution of 10, 20 and 60 m depending on the spectral region (Table 2). In this study, 52 images captured between 5 February 2019 and 11 April 2020 were used. The images were downloaded and processed with the sen2r package [52] in the R software [53].

**Table 2.** Spectral bands of the Sentinel-2 mission. The central wavelength corresponds to the sensors on the Sentinel-2 satellites A and B. Source: Adapted from ESA [42].

| Bands | Spatial Resolution (m) | Central Wavelength (µm) (A;B) |
|---|---|---|
| B1 - Coastal aerosol | 60 | 442.7; 442.2 |
| B2 - Blue | 10 | 492.4; 492.1 * |
| B3 - Green | 10 | 559.8; 559.0 * |
| B4 - Red | 10 | 664.6; 664.9 * |
| B5 - Red Edge | 20 | 704.1; 703.8 * |
| B6 - Red Edge | 20 | 740.5; 739.1 |
| B7 - Red Edge | 20 | 782.8; 779.7 |
| B8 - NIR | 10 | 832.8; 832.9 * |
| B8A - Red Edge | 20 | 864.7; 864.0 |
| B9 - Water vapour | 60 | 945.1; 943.2 * |
| B10 - SWIR - Cirrus | 60 | 1373.5; 1376.9 |
| B11 - SWIR | 20 | 1613.7; 1610.4 * |
| B12 - SWIR | 20 | 2202.4; 2185.7 * |

Note: * band used in the study.

*2.3. Patch Pressure (Pp) Yara Water-Sensor*

The continuous monitoring of the vegetation water status was based on the technique developed by Zimmermann et al. [20], who, using a pressure sensor, recovered the changes in the turgor of the leaf. This technique measures the pressure transfer function of a leaf, called patch pressure (Pp), which corresponds to the variation in the clamping pressure of the sensor as a result of the counter force exerted by the leaf tissue. The Pp is inverse to the turgor pressure, so that the increase in the magnitudes of Pp implies low values of leaf turgor and shows a decrease in the water status of the plant [15].

The turgor pressure sensors (Yara Water-Sensor) were used, made up of two images that were fastened to the plant leaf as indicated in Figure 3. One face had a rectangular cutout that housed a pressure microsensor, which automatically and continuously transmitted a reading every five minutes to a web server, without affecting the development of the plant. To monitor the water status of the vegetation, Yara Water Sensors were installed in five kiwi trees (Table 3), in which the Pp was measured. These sensors were attached to two or three leaves of each tree, which presented similar characteristics (size, color, state of growth). The Pp measurements were made for the 2018 season, between 5 February to 15 May 2019; for the 2019 season, between 12 October 2019 to 11 April 2020.

**Table 3.** Yara Water Sensors installed in each monitored tree.

| Tree | Number of Yara Water Sensors | |
|---|---|---|
| | 2018–2019 | 2019–2020 |
| A05 | 2 | 2 |
| A13 | 2 | 2 |
| A17 | 3 | 1 |
| A31 | 2 | 2 |
| A63 | 2 | 2 |

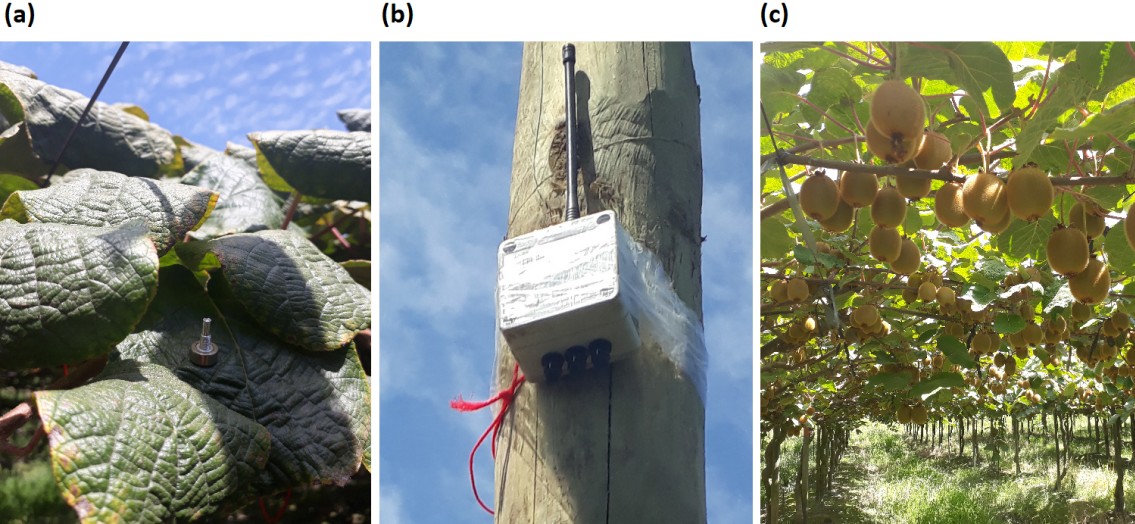

**Figure 3.** (**a**) Yara Water Sensors installed in the field. (**b**) Transmitter that receives the signal from the Yara Water Sensors installed on each leaf of the tree. (**c**) Foliage of the kiwi crop.

### 2.4. Spectrals Vegetation Indices

Nine vegetation indices based on the VNIR and SWIR spectral range derived from the Sentinel-2 satellite were used. Table 4 presents the vegetation indices, among which the enhanced vegetation index (EVI) and the normalized difference infrared index (NDII) are highlighted. This last indicator is found in the literature under various names, associated with a spectral variation in the SWIR and the sensor used. [36] reviewed and determined the names of NDWI, NDII, and NBR for SWIR lengths of (1) 1–1.3 μm, (2) 1.55–1.75 μm, and (3) 2.05–2.45 μm, respectively.

**Table 4.** Evaluated spectral vegetation indices derived from Sentinel-2.

| Wavelength | Vegetation Index | Formula | Reference |
|---|---|---|---|
| VNIR | Enhanced Vegetation Index (EVI) | $\frac{2.5(B8-B4)}{((B8+6B4-7.5B2)+1)}$ | [33] |
| VNIR | Red edge 1 (Rededge1) | $\frac{B5}{B4}$ | [54] |
| VNIR | Leaf Chlorophyll Index (LCI) | $\frac{(B8-B5)}{(B8+B4)}$ | [55] |
| VNIR | Normalized Difference Vegetation Index (NDVI) | $\frac{(B8-B4)}{(B8+B4)}$ | [32] |
| VNIR | Chlorophyll Absorption Ratio Index (CARI) | $\frac{B5}{B4}\frac{\sqrt{(\frac{B5-B3}{150}670+B4+(B3-(\frac{B5-B3}{150}550)))^2}}{(\frac{B5-B3}{150^2}+1)^{0.5}}$ | [56] |
| VNIR-SWIR | Normalized Difference Infrared Index (NDII) | $\frac{(B8-B11)}{(B8+B11)}$ | [38] |
| VNIR-SWIR | Normalized burn ratio Index (NBR) | $\frac{(B9-B12)}{(B9+B12)}$ | [57] |
| VNIR-SWIR | Global Vegetation Moisture Index (GVMI) | $\frac{((B9+0.1)-(B12+0.02))}{((B9+0.1)+(B12+0.02))}$ | [58] |
| VNIR-SWIR | Simple Ratio MIR/NIR Ratio Drought Index (RDI) | $\frac{B12}{B9}$ | [59] |

The vegetation indices were calculated with the *s2_calcindices* function from package *sen2r* [52]. We created a time-series per index, nine in total, having a length of 52 images each and five days frequency. These time-series were procesed in R as stack layer using the package *raster* [60]. We then extracted each index's time-series at the pixel coordinated where the tree was mounted with the Pp. To eliminate data with clouds, from the time-series of NDVI, dates with anomalous values outside the range [0–1] were identified by implementing two criteria: (1) dates with NDVI values less than 0.2 were identified; (2) a visual inspection of the respective RGB composition was made. In Figure 4, the 44 dates selected for the CARI index are presented, which correspond to the index having higher spatial variability through the seasons.

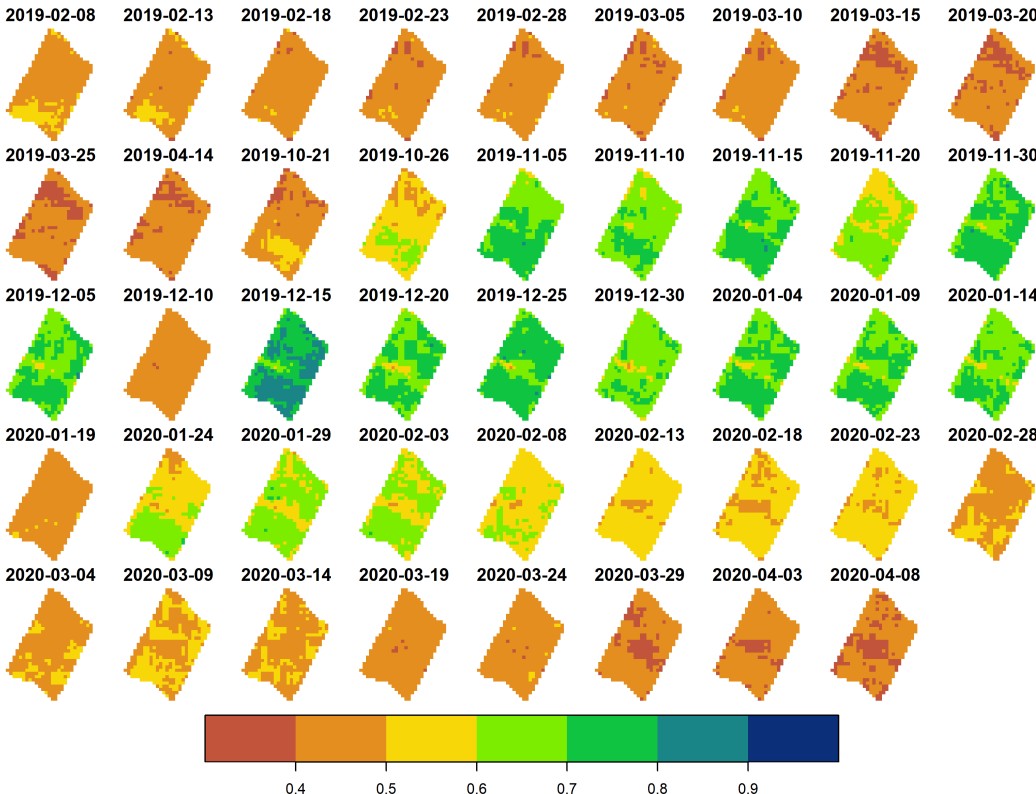

**Figure 4.** CARI time-series for selected dates.

### 2.5. Exploratory Analysis

The Pp data obtained from the different Yara Water Sensors for the 2018–2019 and 2019–2020 seasons were subjected to a cleaning procedure to remove anomalous data. The data from the Yara Water Sensors that had a record lower than 30% were eliminated, and also those records that had at least 3 periods with discontinuities and oscillation of their magnitude of at least 100%. Then, outliers were eliminated for a weekly period, defining a range between the weekly average ± two standard deviations. In addition, the monthly outliers were removed by narrowing a range between the monthly average ± two standard deviations.

Subsequently, the Pp records were summarized daily according to the Sentinel-2 revisit time interval, by calculating the average of Pp every 15 min between 15:00–16:00 UTC, since Sentinel-2 passes over Chilean territory at ~15:30 UTC. Due to the fact that, between the 2018–2019 and 2019–2020 seasons, the kiwi trees lose their leaves, the sensors had to be reinstalled on new leaves from the same tree. Thus, to avoid errors associated with sensor installation, the pressure measurements were considered independent of each other and only the variation of each sensor in each season was considered, which was determined by applying a standardization through the average and standard (Equation (1)) [61].

$$z_i = \frac{x_i - \bar{x}}{\sigma} \tag{1}$$

where $z_i$ value is the standardized value of $x_i$; an element of a data sample with mean $\bar{x}$ and standard deviation $\sigma$.

Regarding the analysis of the vegetation indices, those that presented the lowest correlation between them were selected to avoid redundancy. The selection was based on a correlation matrix based on Pearson's coefficient [62]. Experimentally, the average correlation threshold was determined for each index of $r \leq 0.7$ and $r \leq 0.97$, for indices based on VNIR and VNIR-SWIR, respectively. In order

for the vegetation indices to be comparable with the Pp measurements, the values of each season were standardized with Equation (1).

### 2.6. Correlation Analysis

To relate the Pp measurements with the Spectral indices, the average value of the values in the existing Pp sensors in each tree was used (Table 3). The association of vegetation indices with temporal changes in Pp was evaluated using Pearson's coefficient ($r$) [62] and Spearman's non-parametric coefficient rho ($\rho$) [63]. Additionally, the relationship between both variables was evaluated using a linear regression model, obtaining the metrics of the root mean square error (RMSE) and the coefficient of determination ($r^2$) [64,65].

## 3. Results

### 3.1. Exploratory Analysis

The analysis of the correlation matrix for the vegetation indices (Figure 5) determined that there was a high relationship between the indices constructed with spectral bands at the VNIR-SWIR wavelength, specifically with an $r$ between [0.94–1]. Greater variability was obtained between the indices belonging to the VNIR range with values of $r$ between [0–0.99]. The CARI index stood out for presenting the greatest heterogeneity, which would be explained by the spectral behavior of the green band that is not found in the other indices. The indicators with the lowest correlation that represent the VNIR and VNIR-SWIR spectral range were selected, corresponding to the CARI, EVI, Rededge1, NDII and GVM1 indices, with average $r$ values of 0.17, 0.70, 0.70, 0.96 and 0.97, respectively.

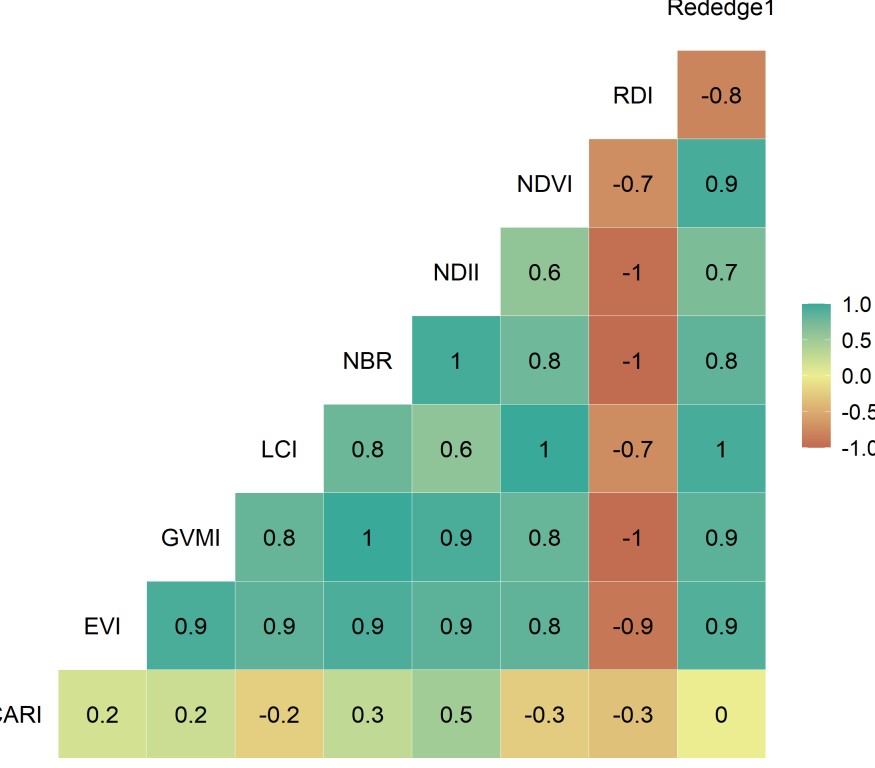

**Figure 5.** Correlation matrix between vegetation indices, value corresponds to Pearson's correlation coefficient (r).

Figure 6b shows the temporal variation in the vegetation indices between each stage of agricultural management (E1, E2, E3 and E4) for each tree. The trend in the vegetation indices in both seasons is in accordance with the phenology and crop management (Figure 2), since the magnitudes of the indices

begin to increase from E1 until reaching the maximum photosynthetic vigor in E2, and from E3 the values in both seasons have a decreasing trend. The spatial variability of the vegetation indices was low, reaching a maximum standard deviation of 0.13, except for CARI, which reached values of 0.43 in the 2018–2019 season. Figure 4 shows the CARI spatio-temporal series, where the spatial variability achieved in the orchard is represented.

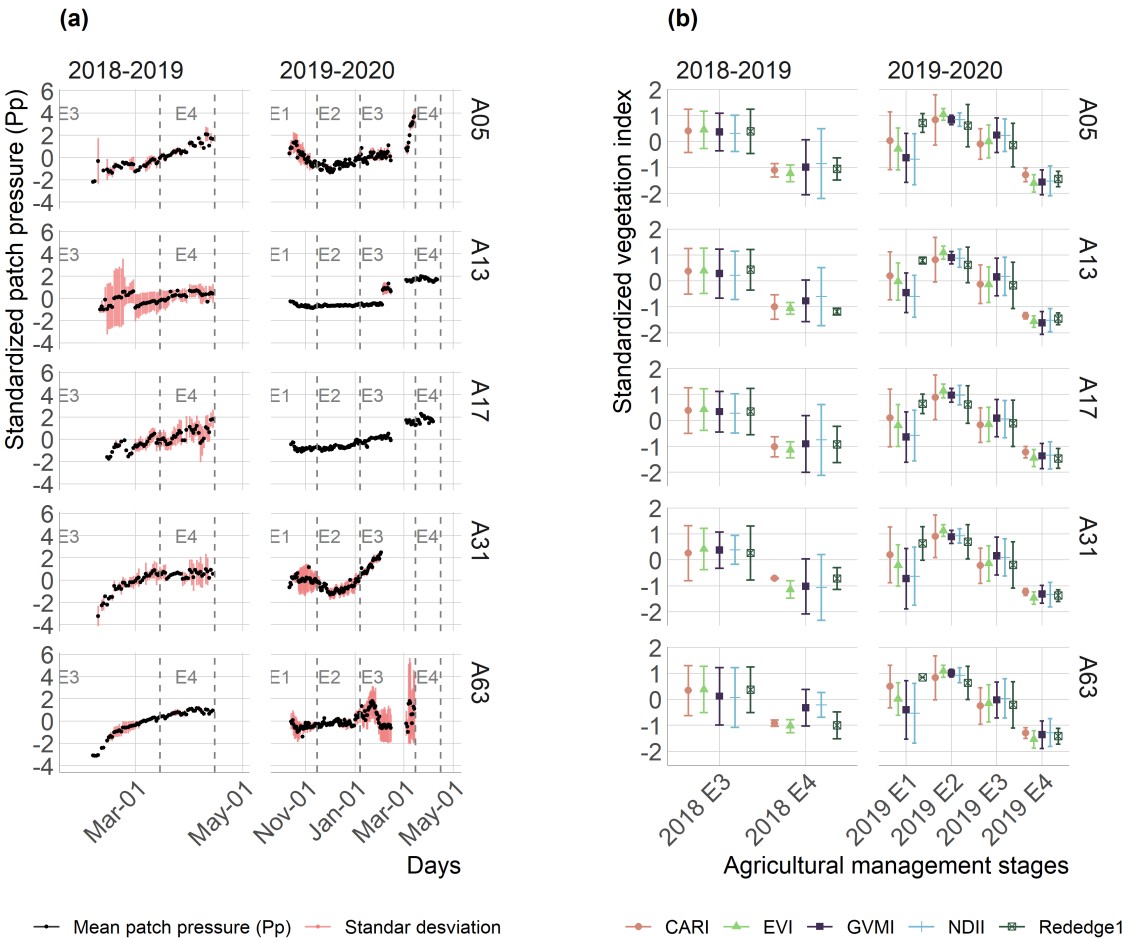

**Figure 6.** (**a**) Average patch pressure (Pp) for each tree (black line) and its standard deviation (red line). (**b**) values of the vegetation indices for each tree grouped for each stage of agricultural management through the average and standard deviation.

Regarding the Pp sensors (Figure 6a), the temporal dynamics present a differentiated behavior between the stages of agricultural management. In E1 and E2, season 2019–2020, the Pp begins with a decrease in its magnitude and then maintains its values, showing a stable behavior over time, which indicates the maintenance of the hydric state of the leaf. On the other hand, in E3 and E4 there is an increase in the magnitudes of Pp, evidenced by the loss of leaf turgor, which is related to a decrease in its hydric state. However, the lack of data on trees for the 2019–2020 season and the variability between sensors limits the reliability of the analysis.

The variability obtained between the sensors of each tree (Figure 6a and Table 5) indicates that it is higher in E3 and E4, with values in the range [0.41, 0.63], followed by E1 and E2, with values of 0.36 and 0.23, respectively. This behavior is related to a differentiated effect of the leaves due to agricultural management, as in the stages where there was greater variability, destructive procedures were carried out on the plant. Specifically, foliage removal tasks were carried out in E1 and E3, and the fruit was harvested in E4. On the contrary, E2 had a non-destructive management corresponding to flower pollination and fruit growth and development (Table 1).

**Table 5.** Average of the standard deviation obtained from Pp in each tree and stage for the 2018–2019 and 2019–2020 seasons.

| Tree | 2018–2019 | | 2019–2020 | | | |
|------|------|------|------|------|------|------|
| | E3 | E4 | E1 | E2 | E3 | E4 |
| A05 | 0.42 | 0.25 | 0.56 | 0.14 | 0.31 | - |
| A13 | 1.34 | 0.64 | 0.11 | 0.04 | 0.12 | 0.05 |
| A17 | 0.38 | 0.90 | - | - | - | - |
| A31 | 0.40 | 0.54 | 0.89 | 0.42 | 0.31 | - |
| A63 | 0.30 | 0.14 | 0.51 | 0.22 | 1.43 | - |
| **Mean** | **0.54** | **0.49** | **0.52** | **0.20** | **0.54** | **0.05** |

### 3.2. Correlation Analysis

The linear regression model used for the Pp and the vegetation indices presented biases in the residuals; therefore, the results of the Pearson correlation coefficient were not included in the analysis and only the Spearman $\rho$ coefficient and the root mean square error (RMSE). Spearman's analysis for the temporal dynamics of Pp and the vegetation indices (Figure 7a) of the 2018–2019 season presented a strong relationship in most of the trees, with EVI and GVMI being the indices that obtained the greatest association with an average value of $\rho$ equal to $-0.88$ and $-0.81$, respectively. CARI, NDII and RedEdge1 reached an average magnitude of $\rho$ equal to $-0.77$, $-0.76$ and $-0.74$ (Figure 7a). Then, in the second season, the magnitude of the association decreased for all the indices, resulting in a $\rho$ equal to $-0.65$, $-0.66$, $-0.55$ and $-0.44$ for Rededge1, EVI, CARI and NDII, respectively. The results indicate that, in the 2019–2020 season, there was a lower relationship between the Pp and the vegetation indices composed of NIR, based on the behaviors in E1 and E2, where the vegetation indices, with the exception of Rededge1, showed an increase in magnitude, while Pp values tended to be constant (Figure 8). It should be noted that the trees that presented the greatest variation between their Yara Water Sensors, A13 and A63 for the 2018–2019 and 2019–2020 seasons, respectively, did not have a significant relationship with the vegetation indices. Figure 7b shows the low Pp prediction capacity of the vegetation indices using a linear model, with an average value of RMSE greater than 0.5 for most of the indices of the first season and greater than 0.7 for the indices of the second season. This indicates an important error in the estimation of Pp, considering that the magnitudes of Pp on average oscillate between $-1$ and 2.

The performance of NDII and GMVI was expected to be higher, as unlike the other indicators, these are composed of a band in SWIR, which is sensitive to the water content of the vegetation. To improve understanding of this result, Figure 9a shows the time-series of the reflectivity of the spectral bands used in the vegetation indices. The reflectivity in red obtained a slight increase from E3, while in SWIR (1610 and 2190 nm) and in Red edge (704 nm) it had low variability, evidencing the lower sensitivity to changes in vegetation. In Figure 9b the average spectral signature of the five kiwi trees is shown for dates at each stage of development (E1-E4). It can be observed that the spectral region that is sensitive to the development of vegetation is in the spectral range 740–1610 nm, corresponding to spectral regions that are not absorbed by healthy vegetation; while regions that are highly absorbed, 442–740 nm and 1610–2202 nm, remain with a low variation. Figure 9a,b indicate that the reflectivity of the vegetation in NIR obtained the highest sensitivity by detecting the phenological behavior of the crop. The behavior of most vegetation indices is mainly explained by NIR.

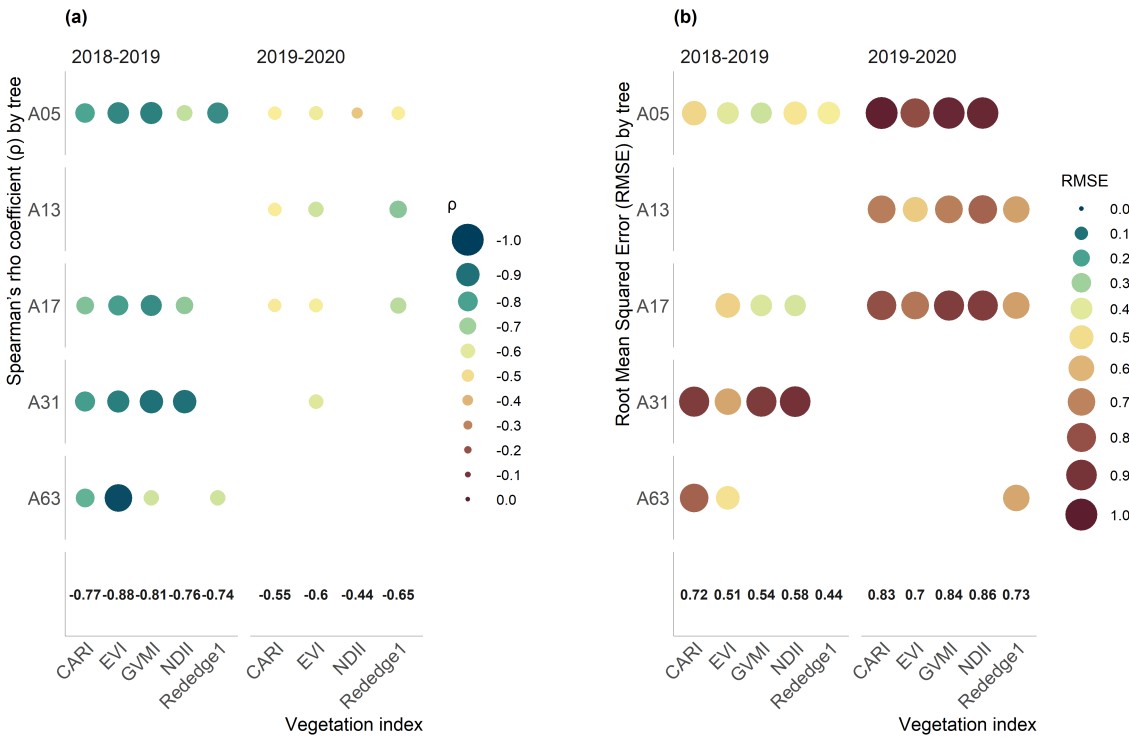

**Figure 7.** (**a**) Spearman's rho correlation coefficient ($\rho$) between Pp and vegetation indices. (**b**) Root mean squared error (RMSE) between Pp and the linear fit of each vegetation index. Both statistical analyzes have a significance of $\rho < 0.05$. The numbers in bold correspond to the average values of $\rho$ for each index.

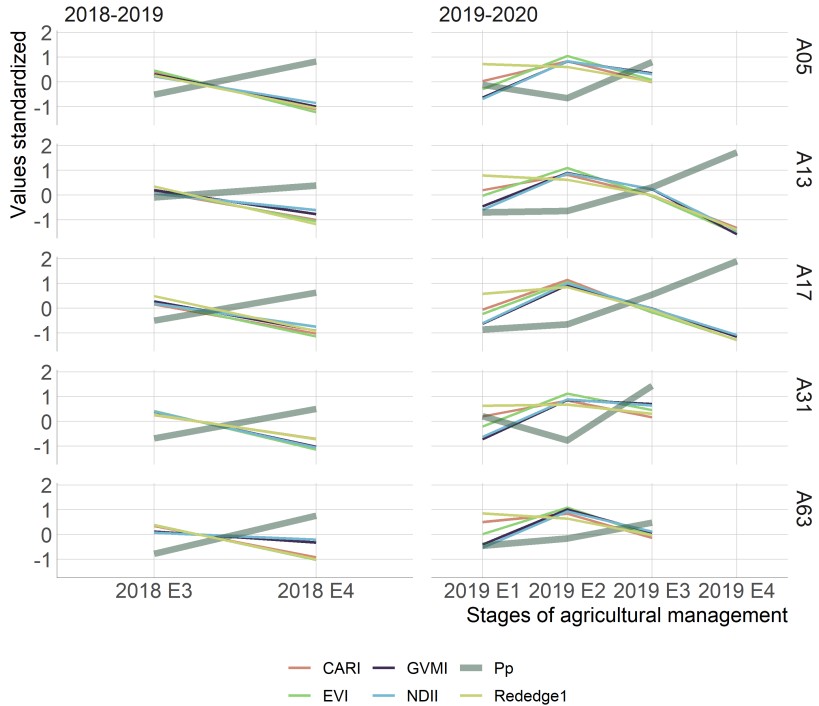

**Figure 8.** Temporal relationship between patch pressure (Pp) and vegetation indices. Average magnitudes per stage are shown.

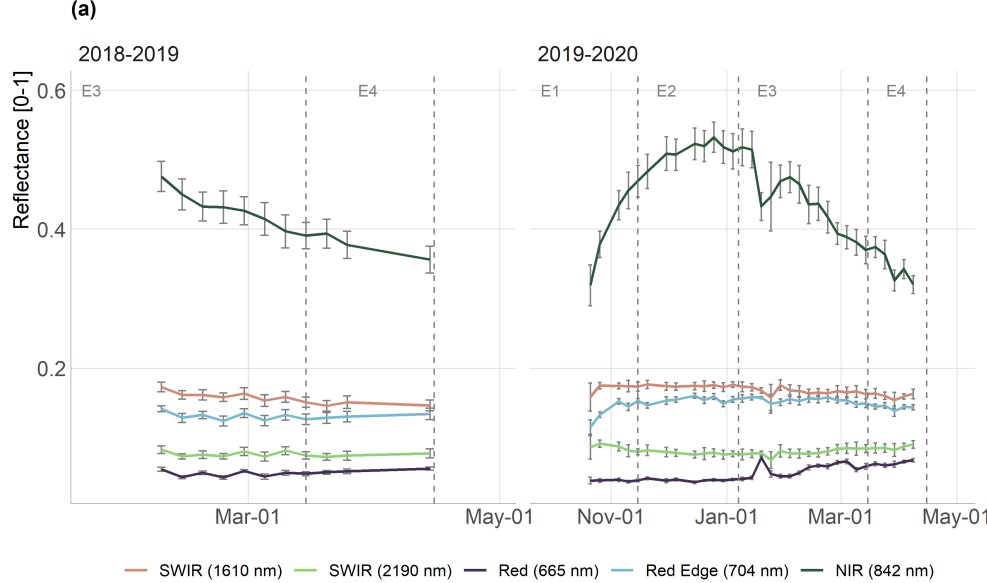

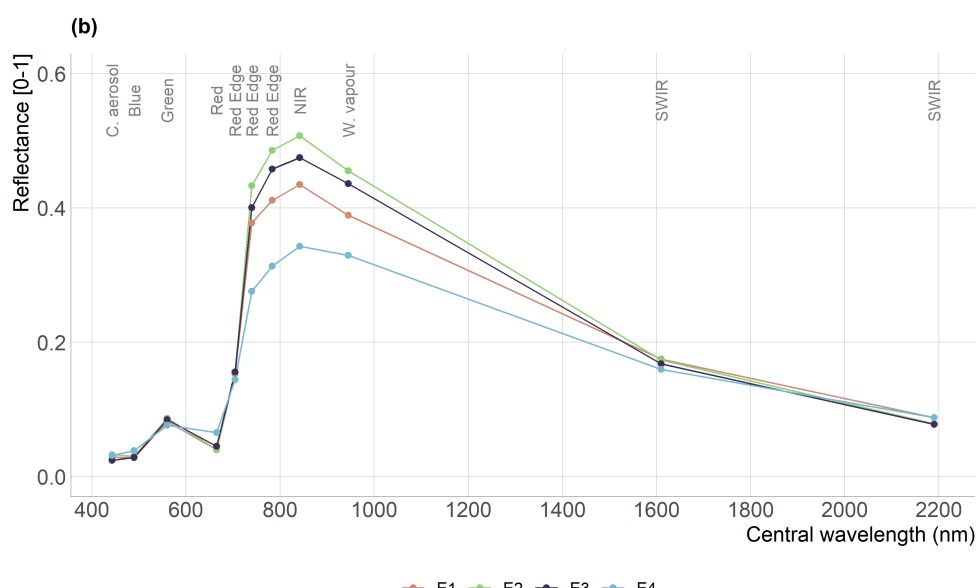

**Figure 9.** (**a**) Reflectivity of the spectral bands of the vegetation indices. The colored line corresponds to the average of the trees and the vertical lines to the standard deviation. (**b**) Average of the spectral signature of the trees for 5 November 2019 (E1), 5 December 2019 (E2), 3 January 2020 (E3) and 3 April 2020 (E4).

## 4. Discussion

Due to the fact that the irrigation supplied was regularly homogeneous and was applied to satisfy the water demand of the crop, the changes in Pp would be explained mainly by the variation in the hydric state of the leaf driven by a synergistic effect between agricultural management, crop phenology and atmospheric demand. Starting at the generation of the fruit, from pollination (E2), the phloem is mobilized from the reserves of the leaves towards the fruit, gradually causing a decrease in leaf turgor [66]. This process is coupled with the greater water losses with the green pruning performed in E3, which reduces the efficiency of the leaf to retain water and does not give in to atmospheric demand.

Finally, the decline in turgor pressure worsens in mid-E4 with the arrival of leaf senescence that generates a decrease in chlorophyll and recycling of foliar nutrients, followed by leaf dehydration [66].

The spectral signature of the kiwi crop, shown Figure 9b, is the average of the reflectance in the kiwi crop pixels in four satellite images corresponding to each management stage's date. Figure 9b indicates that during the four stages of agricultural management, the vegetation maintains a high absorbance in the visible and SWIR lengths, evidencing the behavior of healthy and well-watered vegetation [67], being consistent with the irrigation supply. In this way, it is inferred that wavelengths that kept their reflectivity level constant are more suitable for monitoring vegetation hydric status, being these visible (442–665 nm), red edge (704 nm) and SWIR (1610 and 2190 nm). The Rededge1 index presented the best relationship with Pp in E1-E2, showing a low sensitivity with phenology, specifically with vegetative development, indicating that it is more suitable for monitoring the water status of the vegetation based on the turgor of the leaf. However, as the irrigation supply was applied to satisfy the atmospheric demand, the vegetation did not experience water stress. Therefore, it is necessary to include different irrigation treatments to improve the evaluation of the visible spectral ranges (442–665 nm), red edge (704 nm) and SWIR (1610 and 2190 nm), in order to detect surplus or stress situations in the vegetation and contribute to the development of applications to improve the irrigation strategy of crops. Concerning other studies, Kim et al. [37] evaluated the spectral response of leaves with hyperspectral sensors as their turgor decreased over 24 h, finding that most of the variation in reflectance occurred in the wavelength of SWIR at 1470 nm. Studies that have identified the potentiality of visibility have focused on explaining the water potential. Van Beek et al. [68] correlated spectral indices with the water potential of the stem in Pear Orchards. They found that the higher correlation was in the SWIR (1400–2400 nm), and visible range (500–700 nm) with $R^2$ values of 0.51, and 0.48, respectively, highlighting the advantage of this spectral range to assess the independent hydric status of phenology. Lin et al. [69] used a set of Sentinel-2 bands to predict the water potential of the stem in cotton through random forest, finding that the most important bands correspond to SWIR (1610 nm) and red edge (704 nm).

The vegetation indices made up of NIR showed a close relationship with the vegetative growth of the crop. The lower relationship obtained between Pp and the vegetation indices in the 2019–2020 season compared to the 2018–2019 season, indicates that the relationship between Pp and the vegetation indices composed of NIR is based on the vegetative development of the crop and it is not related to the hydric changes of the crop driven by phenology and agricultural management. This finding was reported by [70] who analyzed the relationship between the reflectance of different spectral ranges and a set of data from temperate and boreal forests of North America. These authors determined that the bands that have the greatest sensitivity with the phenology of the vegetation correspond precisely to the regions that are not used in the photosynthesis process. These are the infrared regions, which are explained by the high correlation obtained between the concentration of canopy nitrogen and the reflectance in the NIR region (800–1400 nm), compared to the low reflectance ratio in the visible region. Further studies should focus on indices that are not sensitive to vegetative development and should relate directly to the vegetation hydric status. In addition to SWIR, spectral bands in the microwave [71,72] and thermal ranges [35,73–76] have been shown to have the qualities to identify variations in surface water content.

It was evidenced that Pp, a variable used as a predictor of the hydric status of the vegetation, is influenced by the agricultural management and phenology of the vegetation. Thus, the analysis should consider irrigation treatments that isolate this response to assess Pp changes driven by the water supply. Finally, in this study, the standardization of Pp was used as a method to remove its magnitude and mainly evaluate its temporal variability. In this way, errors associated with the installation of Yara Water Sensors between different seasons were reduced, which was of great help to represent the behavior of a tree. However, a better analysis of Pp for irrigation applications should include the magnitude to determine Pp ranges that are related to periods of water stress or surplus in the vegetation.

## 5. Conclusions

Nine time-series of vegetation indices in the VNIR and SWIR spectral ranges were used to explain the changes in the hydric state of the leaf of five kiwi trees, recovered through the patch pressure of the Yara Water Sensor. The following conclusions were obtained for the irrigation regimes to which the kiwi orchard was subjected:

- From the nine vegetation indices studies, the CARI index was the one with the lowest temporal correlation ($r = 0.17$) between indices over the kiwifruit trees, which would be explained by the spectral behavior of the green band that is not found in the other indices.
- It was evidenced that continuous measurements of patch pressure (Pp) detected the temporary changes in the leaf's hydric state, which were attributable both to the phenological behavior of the vegetation and agricultural management.
- The crop development highly influenced the performance of the vegetation indices through the season, which explains most of the changes in the water status on the canopy of kiwifruit. Nevertheless, the time-series of vegetation indices that obtained the highest Pp relationship were EVI and Rededge1 for the 2018–2019 and 2019–2020 seasons, respectively.
- The Rededge1 index was less sensitive to vegetative development, and its capacity to monitor the hydric status of the vegetation based on leaf turgor needs to be further investigated.
- Future research needs to address two main issues: (i) to be able to separate the temporal behavior of Vis due to vegetation development to aisle the changes due to the variation of water status in the plant, and (ii) explore with restricted levels of water supply on kiwifruit, which could be implemented with controlled deficit irrigation treatments.

**Author Contributions:** Francisco Zambrano conceive, designed, and performed the experiments; Francisco Zambrano and Alberto Jopia analyzed the data; Waldo Pérez-Martínez, Paulina Vidal-Páez, Julio Molina, and Felipe de la Hoz Mardones contributed with the review of the manuscript and helped to make several major improvements. All authors have read and agreed to the published version of the manuscript.

**Funding:** This research received no external funding.

**Acknowledgments:** This study was financed thanks to the contribution of the FONDECYT project 11190360, of the National Research and Development Agency (ANID), Chile.

**Conflicts of Interest:** The authors declare no conflict of interest. The funding sponsors had no role in the design of the study; in the collection, analyses, or interpretation of data; in the writing of the manuscript, or in the decision to publish the results.

## Abbreviations

The following abbreviations are used in this manuscript:

| | |
|---|---|
| CARI | Chlorophyll Absorption Ratio IndeX |
| $r^2$ | Coefficient of determination |
| EVI | Enhanced Vegetation Index (EVI) |
| ESA | European Space Agency (ESA) |
| ET | Evapotranspiration (ET) |
| GVMI | Global Vegetation Moisture Index (GVMI) |
| LAI | leaf area index (LAI) |
| LCI | Leaf Chlorophyll Index (LCI) |
| NIR | Near infrared (NIR) |
| NBR | Normalized burn ratio Index (NBR) |
| NDII | Normalized Difference Infrared Index (NDII) |
| NDVI | Normalized Difference Vegetation Index (NDVI) |
| NDWI | Normalized Difference Water Index (NDWI) |
| Pp | Patch pressure (Pp) |
| $r$ | Pearson's coefficient |
| RTM | Radiative transfer model (RTM) |
| RMSE | Root mean square error (RMSE) |

| SWIR | Short-wave infrared (SWIR) |
|---|---|
| SIWSI | Shortwave Infrared Water Stress Index |
| RDI | Simple Ratio MIR/NIR Ratio Drought Index |
| SPAC | Soil-plant-atmosphere continuum |
| $\rho$ | Spearman's non-parametric coefficient rho |
| BOA | Bottom of the Atmosphere |
| VNIR | Visible and near infrared |
| WUE | Water use efficiency |

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
