# Peer review of "Time-Series of Vegetation Indices (VNIR/SWIR) Derived from Sentinel-2 (A/B) to Assess Turgor Pressure in Kiwifruit"

_ijgi, doi:10.3390/ijgi9110641_

Round 1
Reviewer 1 Report
The paper is about the use of satellite images from Sentinel-2 A and B to derive vegetation indices in order to assess turgore pressure in kiwifruit.
The topic is interesting and the paper is well written, but some adjustments are needed. I suggest to reconsider it after a major revision.
General comments:
- The authors, starting from the title, write about Time-series but, in my opinion, this aspect is not addressed in an exhaustive way. It is not clear how the authors have treated the identified time series once downloaded;
- The abstract needs to be improved with data supporting the results;
- The introduction is lacking of a part about the state of the art in the use of time series. I suggest to improve this lacking keeping into consideration some of the following articles: Multi Temporal Analysis of Sentinel-2 Imagery for Mapping Forestry Vegetation Types: A Google Earth Engine Approach (doi: 10.1007/978-3-030-48279-4_155), Monitoring gradual ecosystem change using Landsat time series analyses: Case studies in selected forest and rangeland ecosystems (doi: 10.1016/j.rse.2011.06.027), Mapping bamboo with regional phenological characteristics derived from dense Landsat time series using Google Earth Engine. (doi: 10.1080/01431161.2019.1633702);
- In materials and methods section it is not clear how the time series is processed;
- The conclusion chapter has to be improved.
Specific comments:
- Figure 2:To improve the readability of the figure I suggest to rename the management stages starting from january (From January to March E1, from March to April E2, and so on...);
- Figure 2: In the y axis please check the name of the months;
- Figure 2: There is an empty space between April and October. Why? Can the author clarify why this period is not processed?;
- Line 102: Please erase "Each satellite has a time frequency of five days" because it is a repetition;
- Lines 103-105: The authors are saying that the downloaded images have been atmospherically corrected at L2A level. Did the authors download them at level L1C? Why? If not, L2A is already atmospherically corrected, why the use of sen2R package? Could the authors clarify this?;
- Table 4: The EVI formula reported is calibrated for MODIS data. Are the authors sure that all coefficients and bands are correct also for Sentinel-2?;
- Line 130: How did the authors extract the indices values for each tree? Did the authors perform a classification to extract each single tree? Did the authors perform a poligonization of trees? Please clarify;
- Lines 130-132: The authors said that in order to eliminate cloudy images they used a threshold of NDVI value of 0.2. Why did the authors use an NDVI value to discriminate clouds? Why did not the authors mask the image with clouds mask? How did the authors choose 0.2 as threshold value?
- Figure 4: Why the authors are showing only CARI index?
- Line 237: The authors write about the presentation of spectral signature of kiwi crop. How did the authors obtain the signature? Did they choose one image of the time-series? Did they use a mean value of time series? Did they use all the images? Please clarify better how the authors have obtained the signature starting from a time-series;
Author Response
Response to Reviewer 1 Comments
We appreciate the reviewer's comments, which help to improve the manuscript. We have addressed each of your questions point by point. For a better understanding of our review, we have highlighted the text of our response in red. For the answers in which the original manuscript's text was changed, we highlight the original text in color blue and the modificated/new text in green.
General Comments
Point 1: The authors, starting from the title, write about Time-series but, in my opinion, this aspect is not addressed in an exhaustive way. It is not clear how the authors have treated the identified time series once downloaded;

Response 1: We have improved the manuscript in several parts along the text to clearer the isssues related to the time-series and how they were treated.
We changed the text from L5-L6:
We evaluate nine vegetation indices in the VNIR and SWIR regions derived from Sentinel-2 (A/B) satellites to know how much variability in the canopy water status could explain.
By:
We evaluate the time-series of nine vegetation indices in the VNIR and SWIR regions derived from Sentinel-2 (A/B) satellites to know how much variability in the canopy water status could explain.
We changed the text from L129-130:
Vegetation indices were calculated for the 52 available dates. Then, the values of the indices for each tree were extracted.
By:
The vegetation indices were calculated with the s2_calcindices function from package sen2r [52]. We created a time series per index, nine in total, having a length of 52 images each and five days frequency. These time-series were procesed in R as stack layer using the package raster [60]. We then extracted each index's time-series at the pixel coordinated where the tree was mounted with the Pp
Point 2: The abstract needs to be improved with data supporting the results;
Response 2: We have improved the abstract, changing the text from L9-10:
A strong correlation between turgor pressure and vegetation indices was obtained with the Spearman’s rho coefficient (ρ).
by:
A strong Spearman’s ρ correlation between turgor pressure and vegetation indices was obtained, having -0.88 with EVI and -0.81 with GVMI for season 2018-2019; and lower correlation for season 2019-2020 reaching -0.65 with Rededge1 and -0.66 with EVI.
Point 3: The introduction is lacking of a part about the state of the art in the use of time series. I suggest to improve this lacking keeping into consideration some of the following articles: Multi Temporal Analysis of Sentinel-2 Imagery for Mapping Forestry Vegetation Types: A Google Earth Engine Approach (doi: 10.1007/978-3-030-48279-4_155), Monitoring gradual ecosystem change using Landsat time series analyses: Case studies in selected forest and rangeland ecosystems (doi: 10.1016/j.rse.2011.06.027), Mapping bamboo with regional phenological characteristics derived from dense Landsat time series using Google Earth Engine. (doi: 10.1080/01431161.2019.1633702);
Response 3: We have improve the introduction incorporating the suggested research about the use of time-series of satellite data for environmental issues.
We have modified the text on L68-74:
The new multispectral sensor (MSI) attached to the Sentinel-2 (A/B) satellites provides opportunities for vegetation monitoring due to its higher spatial resolution (10/20/60 m), a spectral range of 13 bands from 443 to 2190 nm and a revisit capable of covering the Earth every five days [42]. Simulation studies [43] have been carried out to evaluate the usefulness of Sentinel-2 data to estimate the leaf area index (LAI) and chlorophyll, while its ability to estimate the water content of the crop it has been less investigated.
By:
The new multispectral sensor (MSI) attached to the Sentinel-2 (A/B) satellites provides opportunities for vegetation monitoring due to its higher spatial resolution (10/20/60 m), a spectral range of 13 bands from 443 to 2190 nm, and a revisit capable of covering the Earth every five days [42], allowing the use of high frequent time-series of satellite data for environmental research. Some examples are the use of Google Earth Engine (GEE) and Sentinel-2 to mapping forestry vegetation [43], the use of Landsat data to monitoring ecosystem change [44], and for the mapping of bamboo throughout GEE[45]. Although simulation studies [46] have been carried out to evaluate Sentinel-2 data's usefulness to estimate the leaf area index (LAI) and chlorophyll, its ability to assess the water content of the crop it has been less investigated.
Point 4: In materials and methods section it is not clear how the time series is processed;
Response 4: We addressed this comment in the response to the Point 1 of the reviewer.
Point 5: The conclusion chapter has to be improved.
Response 5: We have improved the conclusion in the following lines
We have modified the text on L275-289:
The Yara Water-sensor patch pressure was used as an indicator of leaf turgor to monitor the water status of five kiwi trees. The following conclusions were obtained for the irrigation regimes to which the kiwi orchard was subjected, which did not consider water restriction:
- It was evidenced that the patch pressure (Pp) detected the temporary changes in the hydric state of the leaf, which were attributable both to the phenological behavior of the vegetation and to agricultural management.
- The vegetation indices that obtained the highest relationship with Pp were EVI and Rededge1 for the 2018-2019 and 2019-2020 season, respectively. However, the performance of the vegetation indices that included the NIR spectral range was influenced by the vegetative development of the crop.
- The Rededge1 index was not sensitive to vegetative development, for which it presented a better performance to monitor the hydric status of the vegetation based on leaf turgor.
By:
Nine time-series of vegetation indices in the VNIR and SWIR spectral ranges were used to explain the changes in the hydric state of the leaf of five kiwi trees, recovered through the patch pressure of the Yara Water-sensor. The following conclusions were obtained for the irrigation regimes to which the kiwi orchard was subjected:
- From the nine vegetation indices studies, the CARI index was the one with the lowest temporal correlation (r =0,17) between indices over the kiwifruit trees, which would be explained by the spectral behavior of the green band that is not found in the other indices.
- It was evidenced that continuous measurements of patch pressure (Pp) detected the temporary changes in the leaf's hydric state, which were attributable both to the phenological behavior of the vegetation and agricultural management.
- The crop development highly influenced the performance of the vegetation indices through the season, which explains most of the changes in the water status on the canopy of kiwifruit. Nevertheless, the time-series of vegetation indices that obtained the highest Pp relationship were EVI and Rededge1 for the 2018-2019 and 2019-2020 season, respectively.
- The Rededge1 index was less sensitive to vegetative development, and its capacity to monitor the hydric status of the vegetation based on leaf turgor needs to be further investigated.
- Future research needs to address two main issues: i) be able to separate the temporal behavior of Vis due to vegetation development to aisle the changes due to the variation of water status in the plant, and ii) explore with restricted levels of water supply on kiwifruit, which could be implemented with controlled deficit irrigation treatments.
Specific Comments
Point 6: Figure 2:To improve the readability of the figure I suggest to rename the management stages starting from january (From January to March E1, from March to April E2, and so on...);
Response 6: Thank you for your comment. Nevertheless, Chile is in the South hemisphere, causing the growing season to start for kiwifruit in August, and the harvest is in May. Because of that, we believe that the stages are best understood in the current status.
Point 7: Figure 2: In the y axis please check the name of the months.
Response 7: was modified
Point 8: Figure 2: There is an empty space between April and October. Why? Can the author clarify why this period is not processed?
Response 8: Thougouth May to Ocober the kiwifruit is without irrigation, because is in dormancy during Autumn and Winter. We have added text to the caption of the Figure 2 to clarify, we changed from:
Figure 2. Variation of crop evapotranspiration and soil moisture content at different depths (15, 35, 55 and 85 cm) in the kiwi orchard for the 2018-2019 and 2019-2020 season. Horizontal lines correspond to the depths of soil moisture of the Sentek probe.
By:
Figure 2. Variation of crop evapotranspiration and soil moisture content at different depths (15, 35, 55, and 85 cm) in the kiwi orchard for the 2018-2019 and 2019-2020 season. Horizontal lines correspond to the depths of soil moisture of the Sentek probe. Blank space between late April and early October corresponds to the period without irrigation.
Point 9: Line 102: Please erase "Each satellite has a time frequency of five days" because it is a repetition;
Response 9: removed
Point 10: Lines 103-105: The authors are saying that the downloaded images have been atmospherically corrected at L2A level. Did the authors download them at level L1C? Why? If not, L2A is already atmospherically corrected, why the use of sen2R package? Could the authors clarify this?;
Response 10: Thanks you for your comment. We have being working for our research with time-series of Sentinel-2 since 2017, some of them we should to atmospherically correct, but those were not used in this manuscript. We have changed the text:
The images were downloaded and atmospherically corrected at L2A level (Bottom of the Atmosphere; BOA) with the sen2r package [49] in the R software [50].
By:
The images were downloaded and processed with the sen2r package [52] in the R software [53].
Point 11: Table 4: The EVI formula reported is calibrated for MODIS data. Are the authors sure that all coefficients and bands are correct also for Sentinel-2?;
Response 11: We used the sen2r package in R to calculate the indices, sen2r get the equation for vegetation indices from the https://www.indexdatabase.de. We have sure we used the bands corresponding with Sentinel-2, which corresponds with EVI equation, (in indexdatabase), which is the same in the manuscript. Nonestanding, the indexdatabese provided other EVI equations (e.g., EVI2), which were not used for this study.
Point 12: Line 130: How did the authors extract the indices values for each tree? Did the authors perform a classification to extract each single tree? Did the authors perform a poligonization of trees? Please clarify;
Response 12: For clariness we have improved the text, and modified were is written:
Then, the values of the indices for each tree were extracted.
By:
We then extracted each index's time-series at the pixel coordinated where the tree was mounted with the Pp.
Point 13: Lines 130-132: The authors said that in order to eliminate cloudy images they used a threshold of NDVI value of 0.2. Why did the authors use an NDVI value to discriminate clouds? Why did not the authors mask the image with clouds mask? How did the authors choose 0.2 as threshold value?
Response 13: To remove the clouds from the dataset, an alternative procedure to the cloud mask was performed, since it did not identify all the clouds that appeared in the images. To clarify this in the manuscript we have changed the following text:
In order to eliminate data with clouds; the dates of the winter and autumn
months were removed with NDVI values lower than 0.2, threshold corresponding to the minimum value that the kiwi crop reached in those months.
By:
To eliminate data with clouds; from the time series of NDVI, dates with anomalous values ​​outside the range [0-1] were identified by implementing two criteria: first 1) dates with NDVI values ​less than 0.2 were identified, and then 2) a visual inspection of the respective RGB composition was made.
Point 14: Figure 4: Why the authors are showing only CARI index?
Response 14: The CARI index was the one with higher spatial variability. To beter understand through the text, we have improved the text in L132-133:
In Figure 4 the 44 dates selected for the CARI index are presented.
Was changed by:
In Figure 4 the 44 dates selected for the CARI index are presented, which correspond to the index having higher spatial variability through the seasons.
Point 15: Line 237: The authors write about the presentation of spectral signature of kiwi crop. How did the authors obtain the signature? Did they choose one image of the time-series? Did they use a mean value of time series? Did they use all the images? Please clarify better how the authors have obtained the signature starting from a time-series;
Response 15: We have modified the text where:
The spectral signature of the kiwi crop shown in Figure 9b, indicates that during the four stages of agricultural management the vegetation maintains a high absorbance in the visible and SWIR lengths, evidencing the behavior of healthy and well-watered vegetation [63], being consistent with the irrigation supply.
By:
The spectral signature of the kiwi crop, shown in Figure 9b, is the average of the reflectance in the kiwi crop pixels in four satellite images corresponding to each management stage's date. Figure 9b indicates that during the four stages of agricultural management, the vegetation maintains a high absorbance in the visible and SWIR lengths, evidencing the behavior of healthy and well-watered vegetation [63], being consistent with the irrigation supply.
Reviewer 2 Report
The manuscript provides the results of investigating vegetation indices based on Sentinel-2 VNIR and SWIR bands to assess the canopy water status on the example of a kiwifruit trees study site in Chile. The manuscript is well structured and the results of satellite monitoring are confirmed by ground-based observations. The recommendation is the following. In the Introduction section the Authors provide a wide range of publications that use vegetation indices to assess water status of crops. In the Discussion section it would be advisable to compare in more detail the results obtained by Authors with the results represented in these publications.
Author Response
Response to Reviewer 2
Comments
We appreciate the reviewer's comments, which help to improve the manuscript. We have addressed each of your questions, point by point. For a better understanding of our review, we have highlighted the text of our response in red. For the answers in which the original manuscript's text was changed, we highlight the original text in color blue and the modificated/new text in green.
Point 1: The manuscript provides the results of investigating vegetation indices based on Sentinel-2 VNIR and SWIR bands to assess the canopy water status on the example of a kiwifruit trees study site in Chile. The manuscript is well structured and the results of satellite monitoring are confirmed by ground-based observations. The recommendation is the following. In the Introduction section the Authors provide a wide range of publications that use vegetation indices to assess water status of crops. In the Discussion section it would be advisable to compare in more detail the results obtained by Authors with the results represented in these publications.
Response 1:
We have improved the text by adding a comparison with other studies, please look at L249
Concerning other studies, Kim et al [37] evaluated the spectral response of leaves with hyperspectral sensors as their turgor decreased during 24 hours, finding that most of the variation in reflectance occurred in the wavelength of SWIR at 1470 nm. Studies that have identified the potentiality of visible have focused on explaining the water potential.Van Beek et al [68] correlated spectral indices with the water potential of the stem in Pear Orchards. They found that the higher correlation was in the SWIR (1400-2400nm), and visible range (500-700 nm) with r2 values of 0.51, and 0.48, respectively, highlighting the advantage of this spectral range to assess the independent hydric status of phenology. Lin et al [69] used a set of sentinel-2 bands to predict the water potential of the stem in cotton through random forest, finding that the most important bands correspond to SWIR (1610 nm) and rededge (704 nm).
Reviewer 3 Report
The manuscript assesses the relationship between vegetation indexes derived from Sentinel-2 with turgor pressure in kiwifruit. The manuscript is well written and needs some adjustment. My recommendation is for your acceptance after minor revision.
Abstract: please describe (i) how many tugor pressure sensors have been installed; (ii) which indexes had the highest correlations. Vegetation indices are expected to represent vegetation rather than water status. Therefore, it is necessary to review the indirect monitoring of water availability in vegetation with the vegetation index in the introduction.
Introduction: Line 48 - Do the authors mean based or remote sensing measurements?
The authors described the different vegetation indices that can be calculated with the reflectances measured by Sentinel-2. However, a review of how these indices can be used to indirectly monitor the availability of water on the surface (soil and vegetation) is lacking.
Figure 1 - The study area would be better described if only using the google image without the EVI. It is not possible to identify the study area by figure 1.
Figure 2 and Figure 4 - Wouldn't that data and images be a result? Why did you only present the images from the CARI and did not present the images from the other indexes?
Author Response
Response to Reviewer 3
Comments
We appreciate the reviewer's comments, which help to improve the manuscript. We have addressed each of your questions, point by point. For a better understanding of our review, we have highlighted the text of our response in red. For the answers in which the original manuscript's text was changed, we highlight the original text in color blue and the modificated/new text in green.
Point 1: please describe (i) how many tugor pressure sensors have been installed; (ii) which indexes had the highest correlations.
Response 1: (i) We modified the text where is stated:
Over the study’s site were installed sensors that continuously measure the leaf’s turgor pressure (Yara Water-Sensor)
By:
Over the study’s site eleven sensors were installed in five trees, which continuously measure the leaf’s turgor pressure (Yara Water-Sensor)
Response 1: (ii) We have improved the abstract, changing the text from L9-10:
A strong correlation between turgor pressure and vegetation indices was obtained with the Spearman’s rho coefficient (ρ).
by:
A strong Spearman’s ρ correlation between turgor pressure and vegetation indices was obtained, having -0.88 with EVI and 0.81 with GVMI for season 2018-2019; and lower correlation for season 2019-2020 reaching -0.65 with Rededge1 and -0.66 with EVI.
Point 2: Vegetation indices are expected to represent vegetation rather than water status. Therefore, it is necessary to review the indirect monitoring of water availability in vegetation with the vegetation index in the introduction.
Response 2:
Point 3: Introduction: Line 48 - Do the authors mean based or remote sensing measurements?
Response 3: It refers to how the reflectance properties of the plan can be affected.
Point 4: The authors described the different vegetation indices that can be calculated with the reflectances measured by Sentinel-2. However, a review of how these indices can be used to indirectly monitor the availability of water on the surface (soil and vegetation) is lacking.
Response 4: Now, we have improved the text addressing your comment.
We change the text in Line 63-66
Hardisky et al. [38] first developed the Normalized Difference Infrared Index (NDII) to infer humidity. Gao [39] proposed the Normalized Difference Water Index (NDWI) using data from Band 2 (841–876 nm) and 5 (1230–1250 nm) from the Moderate Resolution Imaging Spectroradiometer (MODIS) to obtain vegetation water content
By:
Hardisky et al. [38] first developed the Normalized Difference Infrared Index (NDII) to evaluate the spetral respect of Spartina alterniflora to different physiological changes, determining that the wavelength between 1.50-1.75 um presented the best response to the differences in the water content of the leaf. Gao [39] proposed the Normalized Difference Water Index (NDWI) using two channels centered near 0.86 pm and 1.24 pm and determined that NDWI increases as the layers of sheets increase, indicating that NDWi is sensitive to the total amount of water in stacked sheets
Point 5: Figure 1 - The study area would be better described if only using the google image without the EVI. It is not possible to identify the study area by figure 1.
Response 5: We have replaced the Figure by the following
Round 2
Reviewer 1 Report
Dear authors,
thank you for the effort in taking into consideration the comments.
All points have been adequately addressed.